# Morphofunctional and Molecular Assessment of Nutritional Status in Head and Neck Cancer Patients Undergoing Systemic Treatment: Role of Inflammasome in Clinical Nutrition

**DOI:** 10.3390/cancers14030494

**Published:** 2022-01-19

**Authors:** Soraya León-Idougourram, Jesús M. Pérez-Gómez, Concepción Muñoz Jiménez, Fernando L-López, Gregorio Manzano García, María José Molina Puertas, Natalia Herman-Sánchez, Rosario Alonso-Echague, Alfonso Calañas Continente, María Ángeles Gálvez Moreno, Raúl M. Luque, Manuel D. Gahete, Aura D. Herrera-Martínez

**Affiliations:** 1Maimonides Institute for Biomedical Research of Cordoba (IMIBIC), 14004 Cordoba, Spain; soraya.leon.sspa@juntadeandalucia.es (S.L.-I.); jesusmperezgomez@gmail.com (J.M.P.-G.); carri2976@gmail.com (C.M.J.); fernandolopez@imibic.com (F.L.-L.); gmgendocrino@hotmail.com (G.M.G.); cmmerinomjmolina@hotmail.com (M.J.M.P.); nataliahermansan@gmail.com (N.H.-S.); rosario.alonso.echague.sspa@juntadeandalucia.es (R.A.-E.); continente@gmail.com (A.C.C.); mariaa.galvez.sspa@juntadeandalucia.es (M.Á.G.M.); 2Endocrinology and Nutrition Service, Reina Sofia University Hospital, 14004 Cordoba, Spain; 3Department of Cell Biology, Physiology, and Immunology, University of Córdoba, 14004 Cordoba, Spain; 4CIBER Fisiopatología de la Obesidad y Nutrición (CIBERobn), 14004 Cordoba, Spain

**Keywords:** head and neck cancer, malnutrition, sarcopenia, comorbidities

## Abstract

**Simple Summary:**

Malnutrition in patients with head and neck cancer is associated with worse clinical evolution and prognosis. Accurate nutritional assessments allow for early-identification of patients at risk of malnutrition. We aimed to perform a novel morphofunctional nutritional evaluation, including molecular analysis in patients with head and neck cancer who are undergoing systemic treatment. A morphofunctional nutritional assessment includes bioimpedance, anthropometric, ultrasound and biochemical measurements. We observed that malnutrition induces a profound alteration in the gene-expression pattern of inflammasome-machinery components, which are related with clinical nutritional parameters. This molecular analysis should be further studied as potential targets for nutrition-focused treatment strategies in cancer patients.

**Abstract:**

Malnutrition in patients with head and neck cancer is frequent, multifactorial and widely associated with clinical evolution and prognosis. Accurate nutritional assessments allow for early identification of patients at risk of malnutrition in order to start nutritional support and prevent sarcopenia. We aimed to perform a novel morphofunctional nutritional evaluation and explore changes in inflammasome-machinery components in 45 patients with head and neck cancer who are undergoing systemic treatment. To this aim, an epidemiological/clinical/anthropometric/biochemical evaluation was performed. Serum RCP, IL6 and molecular expression of inflammasome-components and inflammatory-associated factors (NOD-like-receptors, inflammasome-activation-components, cytokines and inflammation/apoptosis-related components, cell-cycle and DNA-damage regulators) were evaluated in peripheral-blood mononuclear-cells (PBMCs). Clinical-molecular correlations/associations were analyzed. Coherent and complementary information was obtained in the morphofunctional nutritional assessment of the patients when bioimpedance, anthropometric and ultrasound data were analyzed. These factors were also correlated with different biochemical and molecular parameters, revealing the complementary aspect of the whole evaluation. Serum reactive C protein (RCP) and IL6 were the most reliable parameters for determining patients with decreased standardized phase angle, which is associated with increased mortality in patients with solid malignancies. Several inflammasome-components were dysregulated in patients with malnutrition, decreased phase angle and dependency grade or increased circulating inflammation markers. A molecular fingerprint based on gene-expression of certain inflammasome factors (p27/CCL2/ASC) in PBMCs accurately differentiated patients with and without malnutrition. In conclusion, malnutrition induces a profound alteration in the gene-expression pattern of inflammasome-machinery components in PBMCs. A comprehensive nutritional assessment including novel morphofunctional techniques and molecular markers allows a broad characterization of the nutritional status in cancer patients. Profile of certain inflammasome-components should be further studied as potential targets for nutrition-focused treatment strategies in cancer patients.

## 1. Introduction

Malnutrition may affect 25–50% of patients with head and neck cancers before treatment [1], being severe in about 30% of cases, especially those patients with tumors localized in the oropharynx or the hypopharynx [2]. Malnutrition is related to a higher rate of postsurgical complications, worse treatment response and higher tumor recurrence as well as increases the risk of infections and treatment related toxicity and decreases quality of life/life expectancy [3]. Malnutrition is also related with systemic treatment toxicity and to the tumor itself [1]. Several studies have described malnutrition associated with more treatment interruptions and worse treatment effectiveness [4]. Furthermore, surgery and systemic treatment worsens the nutritional condition due to digestive tract related symptoms, including loss of taste, mucositis, xerostomia, nausea and vomiting [5]. Specifically, loss of fat-free body mass has been proposed as a direct cause for increased mortality and worse prognosis in cancer patients [6], even in patients whose body mass index (BMI) classifies them as overweight, obese or normal [7].

Early detection of patients with head and neck cancers who are at risk for malnutrition is essential for starting early nutrition support in order to prevent or minimize weight loss during treatment [2,7]. The Global Leadership Initiative on Malnutrition (GLIM) has been focused on standardizing the clinical practice of disease related malnutrition (DRM) diagnosis. To this aim, a two-step model was designed based on five top-ranked criteria, including three phenotypic criteria (non-intentioned weight loss, low BMI and reduced muscle mass) and two etiologic criteria (reduced food intake or assimilation, and inflammation or disease burden) [8]. Based on these phenotypic and etiologic criteria, several anthropometric and functional methods have been incorporated into regular clinical evaluation in order to diagnose complex clinical situations such as cachexia and sarcopenia, and to support the management of these pathologies, especially in patients with cancer [9].

In this context, after multi-frequency bioimpedance analysis, phase angle (PA) provides a measurement for energy (electrical) changes, which is related to cell function and the composition of the internal environment. Specifically, nutritional and metabolic changes induce changes in cellular and tissue bioenergy. As a consequence, changes in tissue composition and functionality are produced and might be monitored bioelectrically [9]. 

Additionally, ultrasound has also been described as a helpful tool for providing additional information about the nutritional condition of the patient. It is a new, reliable, easy and non-invasive technique, but muscle and adipose measurements using ultrasound requires validation, especially in terms of morbidity and mortality outcomes [9]. In this context, measurements of the subcutaneous adipose tissue, muscle area and circumference in the femoral area, as well as the possibility of evaluating subcutaneous and visceral fat in the abdominal area, have been described; these measurements are currently the most explored and better standardized ultrasound parameters in clinical practice [10]. Similarly, novel serum markers, specifically inflammation-related markers, that could provide practical, sensitive, specific and reproducible nutritional information during the follow-up are under study [9]. All these novel measurements represent a substantial change in the classical nutritional evaluation of patients, especially in patients with malnutrition and cancer.

Based on this, the use of inflammation-related molecular markers might provide additional, useful information for early diagnosis and prognosis in malnutrition. Specifically, the inflammasome is a multiprotein intracellular complex that detects pathogenic microorganisms and sterile stressors, and that activates highly pro-inflammatory cytokines. Moreover, inflammasome dysregulation has been associated with several inflammatory syndromes, obesity, cancer and autoimmune diseases [11,12].

For all the aforementioned reasons, this study aimed to perform a comprehensive nutritional evaluation of patients with head and neck cancers, who were receiving systemic treatment, and to evaluate the expression and changes in key elements of the inflammasome machinery (i.e., inflammasome components and inflammatory-associated factors) as a complementary tool for the nutritional assessment. To that end, we used peripheral blood mononuclear cells (PBMCs), since disease-associated gene expression patterns are commonly reflected in these cells [13]. Specifically, we analyzed the gene expression levels of four groups of inflammasome machinery components: (1) NLRs or NOD-like receptors; (2) regulators of inflammasome activation; (3) cytokines and inflammation/apoptosis-related components; and (4) cell-cycle and DNA-damage regulators. Additionally, we aimed to explore the putative relations between gene expression levels of these components of the inflammasome machinery with different clinical and biochemical variables, in order to improve the clinical diagnosis of malnutrition and sarcopenia. 

## 2. Materials and Methods

### 2.1. Patients 

This study was approved by the Ethics Committee of the Reina Sofia University Hospital (Cordoba, Spain, approval no 5006), which was conducted in accordance with the Declaration of Helsinki and according to national and international guidelines. This is a prospective open label study, wherein written informed consent was signed by every individual before inclusion into the study. Forty-five patients with head and neck cancer treated with chemo or radiotherapy were included. Clinical records were used to collect full medical history of all patients (demographic and clinical characteristics of patients are summarized in Table 1). All patients were managed following available guidelines and recommendations [14,15]. Body composition was evaluated using a *multi-frequency bioimpedanciometer* (TANITA MC-780MA), which also provided a phase angle (PA) measurement. Standardized phase angle (SPA) was calculated based on age and sex as previously described [16,17]. Waist circumference was measured at minimal expiration. Muscle echography of the rectus femoris muscle of the quadriceps was performed. Adipose tissue, muscle area and circumference were determined at the distal tertium of the thigh [18] using a Midray Z50 Ultrasound. Adipose tissue of the abdomen was measured in the middle point of an imaginary line that binds the belly bottom with the xiphoid appendix [10]. Specifically, total adipose tissue, superficial adipose tissue (superficial and deep-layer) and visceral adipose tissues were measured as previously described [10]. Patients were evaluated when referred to the endocrinology department in our hospital. Blood samples were obtained at the time of evaluation from all patients to isolate and analyze gene-expression levels in PBMCs. 

### 2.2. Blood Sampling and Processing to Isolate PBMCs

Venous blood from all patients was collected in tubes containing EDTA. PBCMs were isolated as previously described [13,19].

### 2.3. RNA Extraction, Quantification and Reverse Transcription 

Total RNA from PBMCs was isolated using Direct-zol RNA kit (Zymo Research, Irvine, CA, USA) following the manufacturer’s instructions, as previously described [13,19,20]. The amount of RNA recovered was determined and its quality assessed by the NanoDrop2000 spectrophotometer (Thermo Fisher, Madrid, Spain). Specifically, all the RNA samples passed the quality controls, being the 260/280 and 230/260 absorbance ratios among 1.8–2.0. As previously described [13,21,22], 1 μg of RNA was reverse transcribed (RT) to cDNA using random hexamer primers with the First Strand Synthesis Kit (Thermo Fisher, Madrid, Spain). RNA from the hepatic HepG2 cell model was Isolated using TRI Reagent (Sigma–Aldrich, Madrid, Spain), followed by dNase treatment, as previously described [19,23,24]. 

### 2.4. Analysis of Components of the Inflammasome Machinery by qPCR Dynamic Array Based on Microfluidic Technology 

A 48.48 Dynamic Array based on microfluidic technology (Fluidigm, San Francisco, CA, USA) was developed and implemented to determine, simultaneously, the expression of 48 transcripts in 45 samples, following the same methods recently described [13,25]. Specific primers for human transcripts of the inflammasome machinery, including NLR-/NOD-like receptors (*n* = 7), regulators of inflammasome activation (*n* = 15), cytokines and inflammation/apoptosis-related components (*n* = 18) as well as cell-cycle and DNA-damage regulators (*n* = 5) have been specifically designed and validated as previously reported [26]. In addition, three housekeeping genes were used. The selection of this panel of genes was based on two main criteria: (1) the relevance of the given inflammasome components and other cell cycle regulators in the inflammatory and apoptotic process, and (2) the demonstrated implication in the inflammatory response in metabolic disorders. 

Preamplification, exonuclease treatment and qPCR dynamic array based on microfluidic technology were implemented following the manufacturer’s instructions using the Biomark System and the Real-Time PCR Analysis Software 3.0 (Fluidigm), as previously described [26,27,28]. The expression level of each transcript was adjusted by a normalization factor (NF) obtained from the expression levels of two different housekeeping genes (beta actin (ACTB) and glyceraldehyde−3-phosphate dehydrogenase (GAPDH)) using Genorm 3.3. This selection was based on the stability of the housekeeping genes analyzed among the experimental groups to be compared, wherein the expression of these two housekeeping genes was not significantly different among groups. 

### 2.5. Statistical Analysis

Between-group comparisons were analyzed by the Mann–Whitney U test (nonparametric data) or the Kruskal–Wallis test (nonparametric data, when we compared more than two groups). Paired analysis was performed by Student’s *t*-test (parametric data) or Wilcoxon test (nonparametric data). Chi-squared test was used to compare categorical data. Statistical analyses were performed using SPSS statistical software version 20, and Graph Pad Prism version 6. Data are expressed as mean ± SEM and percentages. *p*-values < 0.05 were considered statistically significant. Heatmaps and clustering analysis were performed using MetaboAnalyst 4.0 [29]. The inflammasome machinery components that discriminate different groups were selected following two main criteria. First, the VIP score must be higher or equal than 1.5, this value being considered as a significant value in this type of analysis. Second, we chose only those that were sufficient to achieve the best hierarchical clustering in the heatmaps.

## 3. Results

### 3.1. Patient Population and Clinical Evolution

Forty-five patients were evaluated. Most of them were male (62%), with a median age of 64 years-old who presented with tumors of the oral cavity (60%). Over 90% of the evaluated patients received treatment with radiotherapy as monotherapy or in combination with chemotherapy and/or surgery (Table 1). The molecular expression of key inflammasome components was evaluated in PBMCs of all patients, specifically activation components (Figure 1A), NLR/NOD-like receptors (Figure 1B), cytokines, inflammation and apoptosis-related components (Figure 1C), cell cycle and DNA-damage regulators (Figure 1D). The mRNA expression of NLRP1, NLRP3 and NLRP12 was increased in these patients in comparison with other NOD-like receptors (Figure 1B). Among the cytokines, CCL5, CXCR2, TGFB, NFK and IL6R were specially expressed (Figure 1C).

Specific criteria for evaluating malnutrition were used in this study. In this sense, the prevalence of malnutrition in the cohort was evaluated using the GLIM criteria (40%) or a SPA < −1.65 (40%); this last cutoff level was used based on previous reports for increased mortality in patients with cancer and a SPA < −1.65 [16]. No clinical or epidemiological differences were observed between patients with malnutrition using both classification systems except for decreased quality of life measured by the Katz Index of Independence in Activities of Daily Living in patients with decreased SPA (Table 1). Additionally, the anthropometric evaluation differed only in those patients with a SPA < −1.65. Specifically, SPA < −1.65 patients presented with decreased arm and calf circumference, as well as with decreased adipose tissue in the thigh echography independently of the BMI (Table 2).

Biochemical parameters showed decreased albumin levels in patients with malnutrition using GLIM and SPA criteria. Decreased transferrin and total cholesterol was observed only in patients with malnutrition according to the GLIM criteria. In contrast, increased RCP and IL-6 levels were increased when both classification systems were used (Table 3).

### 3.2. Anthropometric-Bioimpedance Analysis Is Complemented with the Functional, Ultrasound Evaluation of Adipose-Muscle Tissue, and with the Molecular Expression of Inflammasome Components

All the evaluated anthropometric, ultrasound, biochemical and BIA parameters were correlated; only significant results were presented in the figures. As expected, BMI was positively correlated with anthropometric parameters, including arms and calf circumference, abdominal, superficial and preperitoneal adipose tissue (Figure 2A). Similar correlations were observed when fat (positive correlations) and lean mass (negative correlations) were evaluated. Additionally, a positive correlation was also observed between lean mass (kg) and arm strength measured by dynamometry while negative correlations were observed between the percentage of water and ultrasound evaluation of fat tissue (Figure 2A).

Furthermore, BMI, body cell mass (BCMe), extracellular body cell mass (ECMe), lean muscle mass, water and bone mass (kg) were positively correlated with hemoglobin serum levels and negatively correlated with serum IL-8 values. 25-hydroxi (OH) vitamin D was negatively correlated with BMI, and serum HDL-cholesterol levels were correlated with ECMe, water and cone mass (Figure 2B). Regarding bioimpedance and molecular analysis, BCMe and AIM2 were positively correlated; fat mass was positively with IKKA, FN1 and CXCR2; percentage of lean mass and water (%) negatively correlated with IKKA and CCL8, while water and one mass (kg) were positively correlated with the mRNA levels of AIM2 (Figure 2B).

Finally, anthropometric and functional parameters correlated with some biochemical serum values of hemoglobin and visceral proteins (albumin, prealbumin); specifically, dynamometry correlated only with hemoglobin, prealbumin and mRNA levels of TGFB (Figure 2C). 

### 3.3. Adipose and Muscle Tissue Evaluation Using Ultrasound Provides Additional Information about Nutritional Status in Patients with Head and Neck Cancers

Muscle echography of the right thigh was performed in all patients. Muscle area was positively correlated with albumin and prealbumin serum levels (Figure 3A). Positive correlations with several inflammasome components (NLRC4, ASC, caspase 5 and DCR2) were observed, as well as some negative correlations (CXCL3 and CCL2) (Figure 3A). In contrast, the circumference of the rectus femoralis was positively correlated only with NLRC4 and p19. The overlying adipose tissue was negatively correlated with serum ferritin levels and AIM2 expression, and a positive correlation with IKKA was also observed (*p* < 0.05; Figure 3A).

The abdominal ultrasound evaluation also correlated with biochemical and molecular parameters. Specifically, total adipose tissue of the abdomen was negatively correlated with the mRNA levels of NLRC4 (Figure 3B). Preperitoneal fat positively correlated with prealbumin levels and negatively correlated with serum levels of RCP, IL6, IL8 and serotonin and the mRNA expression levels of P2X7, NKF and MAP14 (Figure 3B). In contrast, superficial adipose tissue positively correlated with the molecular expression of IL18R and IFN, and negatively with TLR4 expression (Figure 3B).

### 3.4. Inflammasome Components Are Correlated with Biochemical Nutritional Parameters in Patients with Head and Neck Cancer

NLR-/NOD-like receptors (NLRP3, NLRP7, NLRC4 and BIRC1) were correlated with serum lymphocytes, prealbumin, HDL, LDL triglycerides, zinc and IL8 (Figure 4). Inflammasome activation components (AIM2, IFI16, ASC, IL1, IL18R, IL1RA, caspase 4, IL6R, TLR4 and P2X7) positively correlated with hemoglobin, albumin, total-, HDL-, LDL-cholesterol, RCP, IL8 and 25-OH vitamin D serum levels (Figure 4). These molecular components negatively correlated with prealbumin, serotonin, zinc and triglycerides (Figure 4). The group of cytokines and inflammation/apoptosis related components (NFK, CCL2, CCL5, CCL7 and CXCR1) positively correlated with lymphocytes, HDL-cholesterol, IL6, IL8 and albumin, and negatively with LDL-cholesterol, albumin and IL8 (Figure 4). Finally, cell-cycle and DNA-damage regulators (p16, ATM) were positively correlated with serotonin and albumin respectively (Figure 4). 

### 3.5. Decreased BMI and Malnutrition Are Associated with Clinical and Molecular Variables in Patients with Head and Neck Cancers

Normal BMI ranges are 22.1–24.9 kg/m^2^ in patients older than 65 years-old, and 18.5–24.9 in adults younger than 65 years-old according to specific consensus documents for the elderly [30]. According to this classification, as expected, patients with decreased BMI (adjusted by age) presented with decreased BCMe, ECMe, phase angle, abdominal circumference, arm and calf circumference, fat and lean body mass (*p* < 0.05) (Figure 5A). All these associations reveal the appropriate classification and evaluation of the cohort of patients included in the present study. Regarding the ultrasound evaluation, decreased adipose tissue in the thigh and preperitoneal fat were also observed, despite increased superficial adipose tissue in the abdomen. In these patients, decreased serum prealbumin levels were observed as well as increased mRNA expression levels of IL6R and P2X7 (Figure 5A). Additionally, patients with malnutrition (according to the GLIM criteria) presented with decreased phase angle (PA) and standardized PA (SPA), decreased serum levels of albumin, transferrin and total cholesterol, accompanied by increased serum RCP and IL6 (Figure 5B). The expression of the cell-cycle and DNA-damage regulators p27 was significantly decreased (Figure 5B).

### 3.6. PA Is Associated with Serum IL6 and Dependency in Patients with Head and Neck Cancers 

A decreased PA for the Caucasian population (PA < 9), according to the standardized software of the TANITA bioinmpedanciometer, was associated with lower BMI, calf circumference and adipose tissue thickness in the thigh (Figure 6A). The SPA of −1.65 also was associated with increased serum IL6 and decreased femoral adipose tissue (Figure 6B). Those patients that presented any level of dependency had lower phase angle, calf circumference and prealbumin levels (*p* < 0.05), also accompanied by decreased expression of AIM2 and p21 (Figure 6C).

### 3.7. Serum Inflammation Markers as Indicators of Nutritional Status in Patients with Head and Neck Cancers

Increased RCP levels (RCP > 10 mg/L) were associated with decreased BMI, preperitoneal fat and prealbumin levels, as well with increased serum IL6 and IL8 levels (Figure 7A). Additionally, patients that presented with increased serum IL6 (IL6 > 4.4 pg/mL) had lower PA, total, subcutaneous and deep adipose tissue in the abdomen, as well as decreased preperitoneal fat (Figure 7B). Regarding the biochemical parameters, decreased serum IL6 was associated with lower albumin and total cholesterol levels and increased RCP levels (Figure 7B). Considering the molecular expression of inflammasome components, increased expression of IFI16 and decreased IL1RA and SIRT1 expression were associated with decreased serum IL6 (Figure 7B).

### 3.8. Inflammasome Components Are Correlated to Standardized PA and Malnutrition

Serum levels of IL6 and RCP, as well as the expression pattern of all the evaluated inflammasome components in patients with SPA < and >−1.65 were submitted to clustering analysis. Specific VIP scores of the evaluated inflammasome components were calculated (Figure 8A). According to this bioinformatic analysis, serum IL6, RCP and the molecular expression of ASC, CCL2, IL1R, p27, CXCL3 and NLRP12 were the components that better discriminated both group of patients and are depicted in Figure 8B by a heat map.

A similar analysis was performed after classifying patients according to the GLIM criteria. Specific VIP scores of the evaluated inflammasome components were calculated (Figure 9A). According to this bioinformatic analysis, the molecular expression of p27, ASC, CCL2 and NLRP12 were the components that better discriminated both group of patients (Figure 9B).

## 4. Discussion

This study was aimed at performing a comprehensive nutritional evaluation of patients with head and neck cancer, who were receiving systemic treatment. Additionally, we evaluated the utility of the expression profile of key elements of the inflammasome machinery in the PBMCs of these patients, as a complementary tool for their nutritional assessment, in order to improve the clinical diagnosis of malnutrition and sarcopenia. To the best of our knowledge, this is the first report demonstrating that the expression of specific and relevant inflammasome components is drastically dysregulated in patients with malnutrition and head and neck cancers, and that a comprehensive nutritional and molecular assessment, including novel morphofunctional techniques and molecular markers can be really useful to obtain a broad characterization of the nutritional status in patients with head and neck cancers. 

Classic imaging techniques for body composition evaluation, such as dual-energy X-ray absorptiometry (DXA), computed tomography and magnetic resonance imaging, are considered “gold standards” for body composition measurements, but their regular application in clinical practice is challenging and not routinely used [31,32]. For this reason, novel approaches have been described in order to provide similar clinical information using easier, cheaper, faster and more reliable methods. In this sense, BIA measures the electrical impedance of body tissues and has been used to assess fluid volumes, total body water, body cell mass and fat-free body mass. The impedance of tissues is comprised of resistance and reactance; body fluids represent the resistive component, whereas the cell membranes represent the reactive component [33]. Results might be altered in cases of edema or dehydration; for that reason, additional body composition methods should be evaluated. Our results demonstrate that body composition using BIA is widely correlated, as expected, with classical anthropometric parameters (calf, arm and abdominal circumference) [34,35], and additionally with ultrasound parameters of muscle mass and adipose tissue, especially in the abdominal area, suggesting the reliability of our measurements, since standardization for the measurement is lacking and results that are highly dependent on operator proficiency [36].

Moreover, several biochemical parameters, including albumin, transferrin, prealbumin lymphocytes and cholesterol are general markers commonly used in nutritional evaluation. These parameters have been correlated with whole-body protein, energy status or nutrient balance [9]. Albumin has been associated with morbidity and mortality in different clinical scenarios [37,38]. In addition, other proteins such as prealbumin and transferrin have been used for evaluating early-nutritional recovery due to their half-life [39]. Total lymphocyte count and low cholesterol levels have been associated with energy restriction [40,41]. These biomarkers are exposed to many interferences from inflammatory processes since many of them behave similarly to acute phase reactants [38], which limits their application in several cases and should be interpreted carefully. Despite this, we observed in our cohort several correlations with BIA measurements (including fat and lean mass), anthropometric values (circumferences), functional tests (dynamometry) and ultrasound values.

Furthermore, when specific groups of patients were evaluated in our cohort (decreased BMI adjusted by age, malnutrition according to the GLIM criteria, decreased PA or SPA and dependency), expected associations with body composition, anthropometric measurements, dynamometry and adipose/muscle measurements were observed, suggesting the reliability of the nutritional evaluation in our cohort and the accuracy of the used classifications. Importantly, this novel clinical evaluation still requires standardization and might be affected by instruments and inter-individual variations. 

Recently, the American Society for Parenteral and Enteral Nutrition published a statement recognizing albumin and prealbumin as important factors that correlate the risk for adverse outcomes in patients, but should not be used for diagnosing protein-energy malnutrition [42]. This statement is based on the fact that there is an association between inflammation and malnutrition, but not between malnutrition and visceral-protein levels [42]. This fact suggests the necessity of developing novel complementary evaluation factors for nutritional status. In this context, circulating IL6 and RCP levels have been described as sarcopenia-associated inflammatory markers, and are included in the GLIM criteria for the diagnosis of malnutrition [8,43]. In our cohort, RCP and IL6 were correlated among participants and associated with other nutritional parameters, especially with PA values and decreased adipose tissue measurements (in both, the abdomen and in the thigh). Indeed, both markers represented the most reliable inflammation-related parameters that discriminated patients with SPA < −1.65, which is associated with increased mortality in cancer [16]. Despite this, IL6 and CRP are not specific nutritional markers, thus, their use in nutrition should be in combination with other clinical or biochemical factors [8]. For this reason, we performed a comprehensive molecular analysis of inflammasome-related components, in order to determine the best combination of markers that could help to improve the nutritional evaluation of cancer patients.

There is a well-known link between diseases, inflammation and malnutrition, specifically with the loss of lean body mass, muscle size, cellularity and leg muscle protein [44]. However, available information about inflammasome machinery in nutrition is limited. Specifically, previous publications have described that weight loss due to calorie restriction reduces oxidative stress and adipose tissue inflammasome activation in mice and humans with type-2 diabetes [45], which is reflected in reductions of CRP and TNFα levels [46]. Indeed, calorie restriction upregulates pathways that inhibit activation of some inflammasome components, including NLRP3. NLRP3 has been widely described in obesity, and it is upregulated in the subcutaneous and visceral adipose tissue from patients with obesity [47,48,49,50,51]. Moreover, NLRP3 expression has been also positively correlated with increased BMI, insulin-resistance and negatively correlated to adiponectin levels [47,49]. In our cohort, NLRP3 was not associated with any of the used classifications in our cohort, even in patients with normal or increased BMI. These findings suggest that the modulation of NLRP3 due to calorie restriction occurs specifically in patients with obesity and when calorie restriction avoids malnutrition [52]. Thus, from a general point of view, we should expect decreased inflammasome activation in patients with malnutrition due to weight loss, accompanied by the insulin resistance of cancer patients [53].

Interestingly, we found that the expression of CDKN1B (p27), a cell-cycle and DNA damage regulator, was significantly decreased in our cohort of patients with malnutrition and was the most reliable marker for differentiating malnutrition in head and neck cancer patients. Nutrients act as growth factors; their presence allows cells to grow and proliferate [54], and muscle cell proliferation is inhibited in malnourished patients [55], which might explain the significant changes in the expression of this molecular factor in our study group.

Moreover, we found that the cytokine CCL2 was negatively correlated with muscle area, and it was also correlated with lipid markers (HDL and triglycerides) in our study. This inflammasome component was also highly accurate for discriminating patients with malnutrition and SPA < −1.65. CCL2 is known to be increased after muscle injury and is necessary for muscular recovery and tissue regeneration [56]. In this context, patients with cancer are characterized by a catabolic state, with increased skeletal-muscle-protein turnover [57], which might explain the significant changes in CCL2 expression found in our study.

Additionally, the inflammasome activation component ASC was positively correlated with albumin levels and femoral muscle area in our study. In fact, after serum IL6 and RCP levels, ASC was the most accurate marker for classifying patients according to their SPA. In humans, increased levels of ASC have been observed in abnormal and dystrophic muscle tissue, and these findings have also been confirmed in mouse models, suggesting that muscle cells can actively participate in inflammasome formation, using ASC as a key factor [58]. As with CCL2, significant changes in muscle composition are observed in patients with cancer and malnutrition, suggesting an important relation between muscle homeostasis, inflammation and inflammasome components. 

Finally, the NOD-like receptor that was more likely involved in the clinical differentiation between malnutrition and decreased SPA in our cohort was NLRP12. NLRP12 is recognized as a potent mitigator of inflammation, it is a checkpoint of obesity, restrains high fat diet-induced inflammation and regulates bowel inflammation [59,60]. NLRP12 has been also associated with changes in gut microbiota [60]. Recent publications suggest that cancer therapies could impact gut microbiota composition and functions [61]; this microbiota modulation might improve immune response and clinical evolution of these patients [61,62], but the specific role of this inflammasome component in nutrition should be further investigated since no current publication, to the best of our knowledge, links this receptor with nutritional status or evolution.

## 5. Conclusions

In conclusion, our results reveal novel conceptual and functional pathways in the nutrition field with potential clinical implications, by demonstrating for the first time that a morphofunctional and molecular nutritional assessment in patients with head and neck cancer could be useful to obtain a broad characterization of the nutritional status in these patients. Additionally, a clear dysregulation of key components of the inflammasome machinery (especially p27, CCL2 and ASC) in these patients has been observed, which may be closely related to the diagnosis of malnutrition and its clinical consequences. Specifically, our data point out that these specific components of the inflammasome machinery in combination with serum inflammation markers might play an important role as indicators for predicting malnutrition and related comorbidities, and that they might represent putative targets for reversing malnutrition in cancer patients. Additionally, our study provides solid, convincing evidence demonstrating that some components of the inflammasome machinery are associated and might play a critical physio-pathological role in the nutritional evaluation of cancer patients, offering a clinically relevant opportunity for novel targets that should be evaluated.

## Figures and Tables

**Figure 1 cancers-14-00494-f001:**
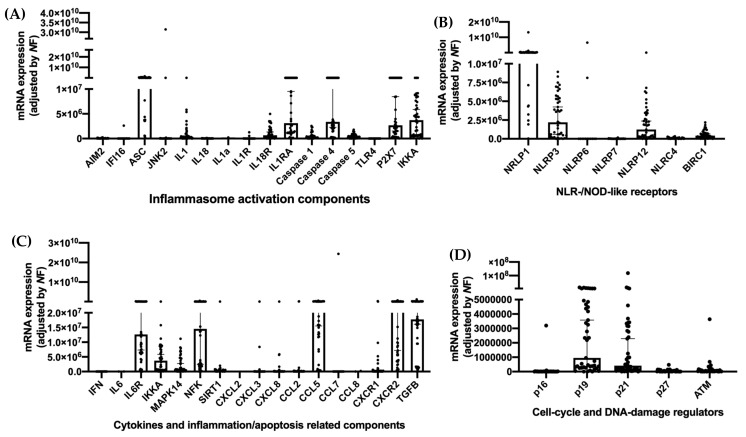
mRNA expression of key components of the inflammasome system. (**A**) Activation components; (**B**) NLR/NOD-like receptors; (**C**) Cytokines and inflammation/apoptosis related components; (**D**) Cell-cycle and DNA damage receptors.

**Figure 2 cancers-14-00494-f002:**
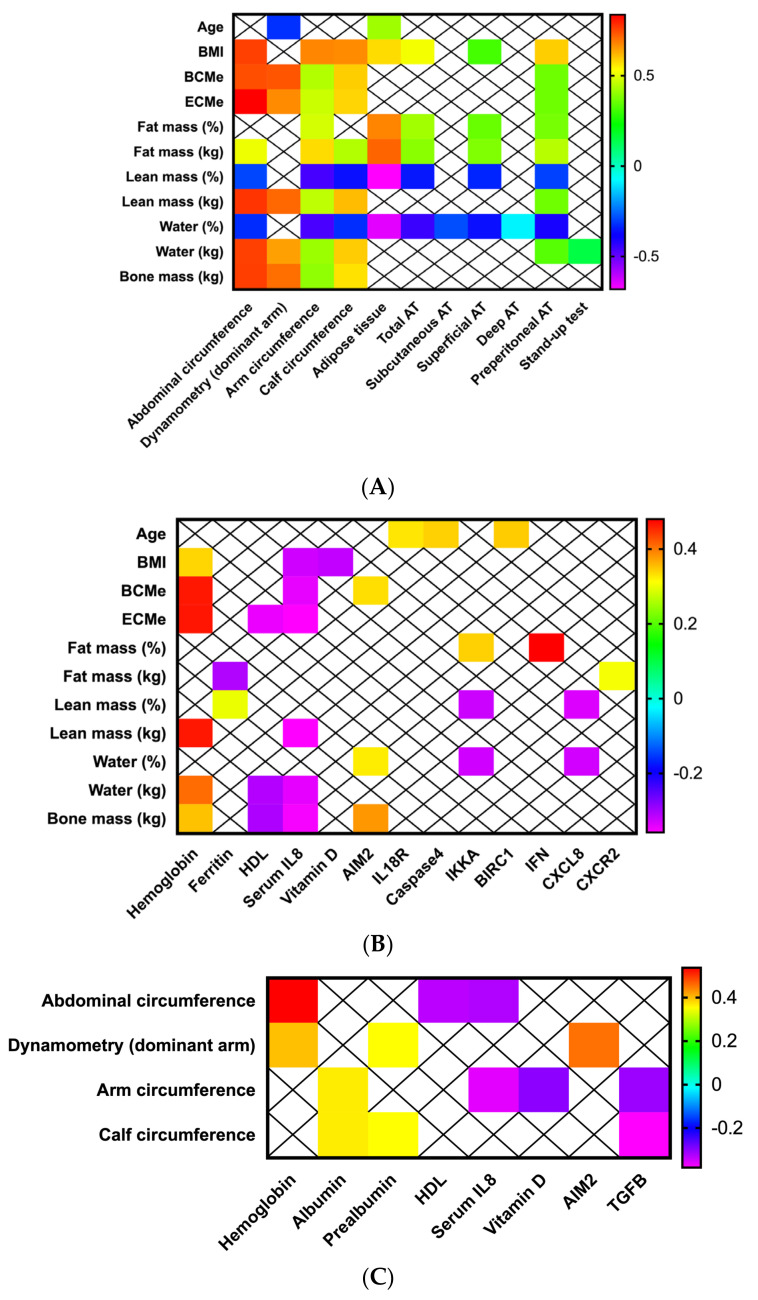
Morphofunctional nutritional assessment in patients with head and neck cancers. Correlations between (**A**) Bioimpedance analysis, anthropometric evaluation, muscle and adipose tissue echography; (**B**) Bioimpedance analysis, biochemical and molecular parameters; (**C**) Anthropometric, functional, biochemical and molecular parameters.

**Figure 3 cancers-14-00494-f003:**
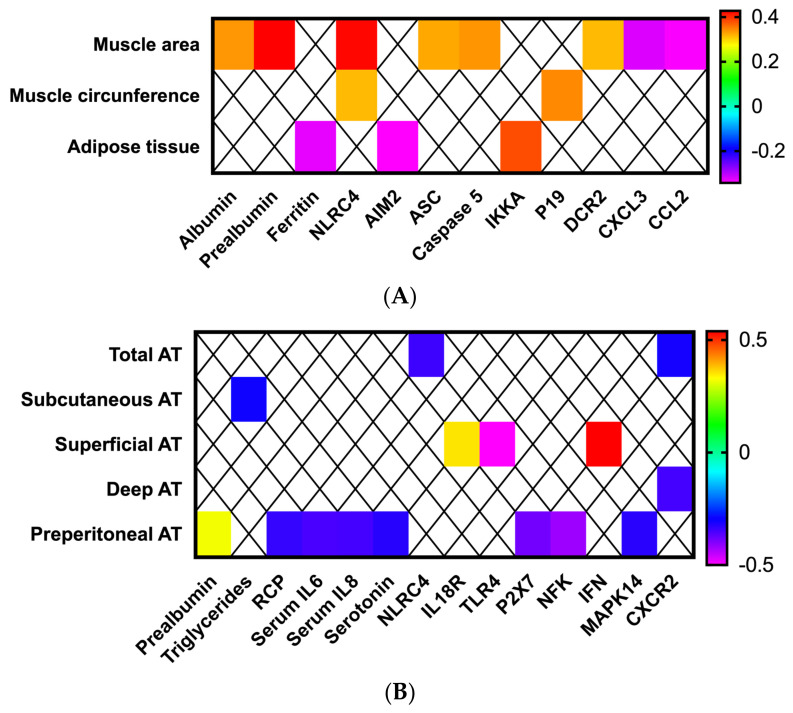
Morphofunctional nutritional assessment in patients with head and neck cancers. Correlations between (**A**) muscle echography, biochemical and molecular parameters, (**B**) adipose tissue echography, biochemical and molecular parameters.

**Figure 4 cancers-14-00494-f004:**
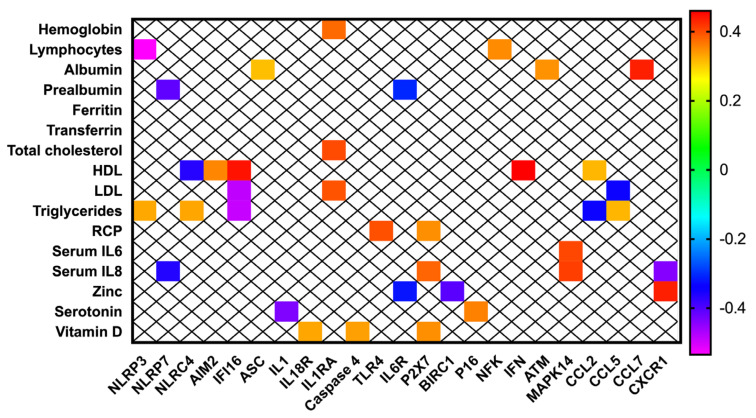
Correlations between biochemical and molecular parameters of the nutritional evaluation in patients with head and neck cancers.

**Figure 5 cancers-14-00494-f005:**
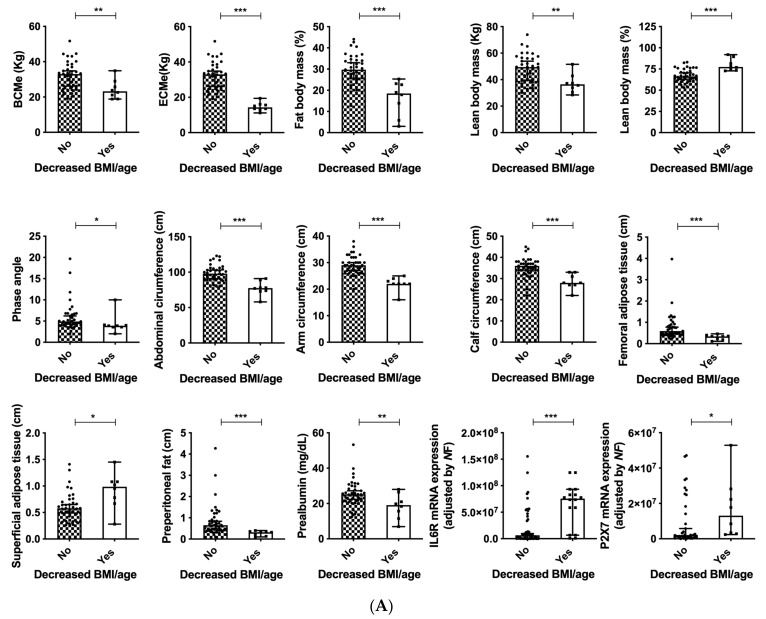
(**A**) Clinical and molecular associations in patients with decreased BMI (adjusted by age) and head and neck cancers; (**B**) Clinical associations in patients with head and neck cancers and malnutrition. Legend: *: *p* < 0.05; **: *p* < 0.01; *** *p* < 0.001.

**Figure 6 cancers-14-00494-f006:**
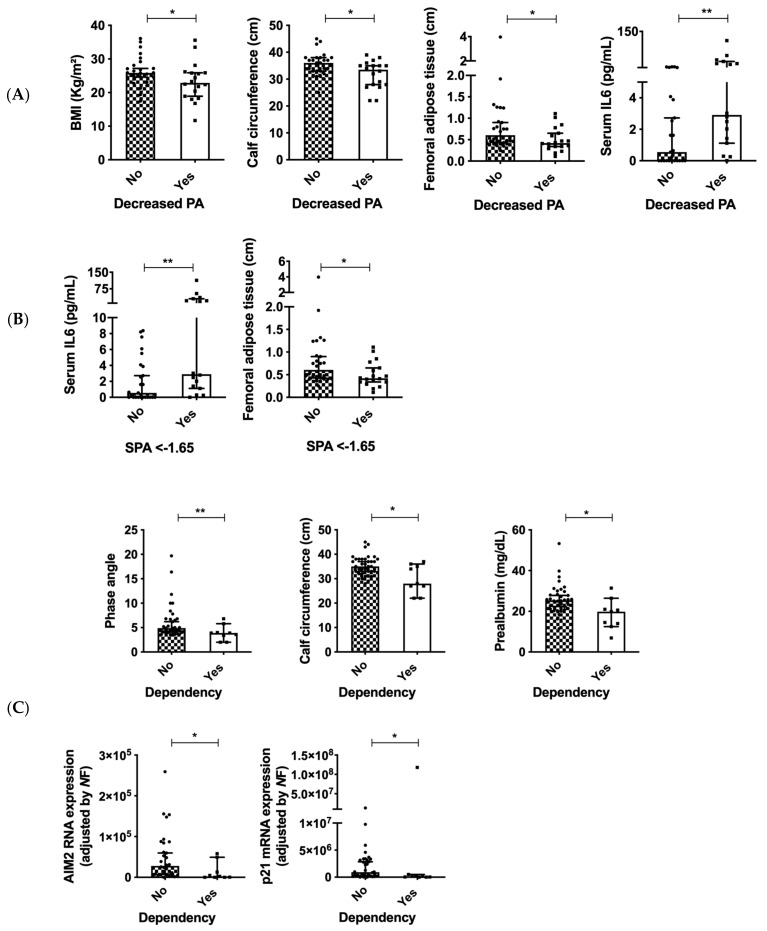
(**A**) Clinical and molecular associations in patients with decreased PA, SPA (**B**) and any level of dependency (**C**) in patients with head and neck cancers. Legend: *: *p* < 0.05; **: *p* < 0.01.

**Figure 7 cancers-14-00494-f007:**
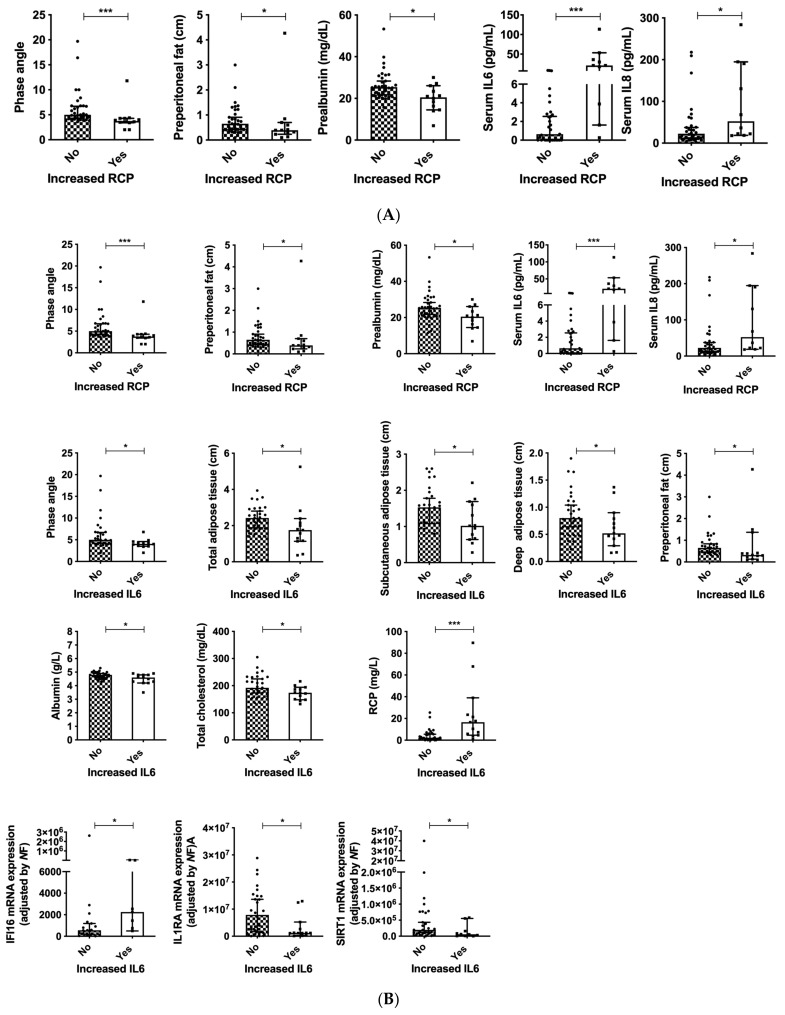
Clinical and biochemical associations in patients with increased serum inflammation markers and head and neck cancers. (**A**) Increased RCP; (**B**) Increased serum IL6. Legend: *: *p* < 0.05; *** *p* < 0.001.

**Figure 8 cancers-14-00494-f008:**
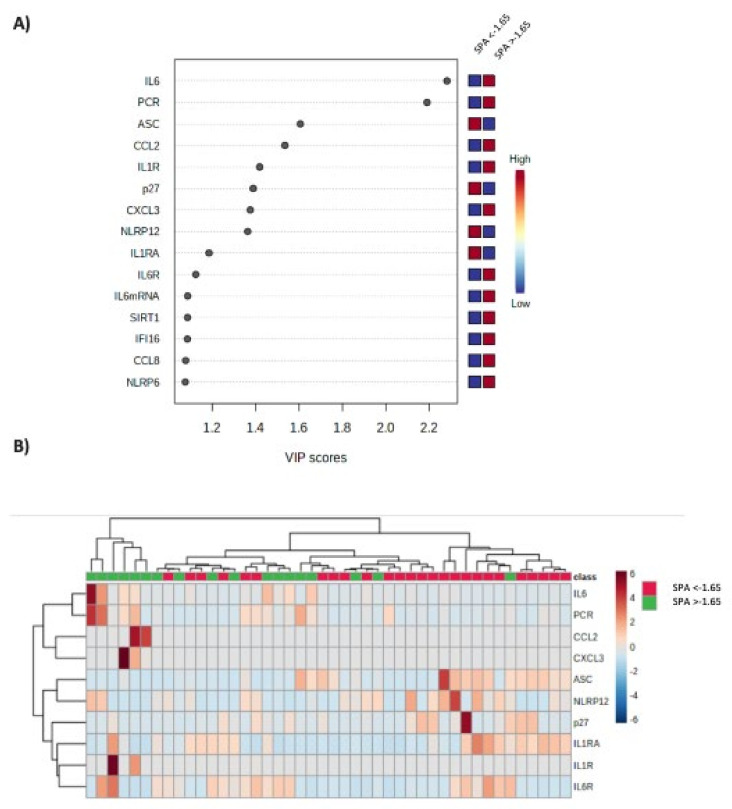
(**A**) VIP scores that summarize the contribution of the evaluated inflammasome components to discriminate patients with SPA <−1.65. (**B**) Heatmap obtained with the expression levels of the components that better discriminate between patients with SPA <−1.65 and >−1.65 using bioinformatics analysis of clustering.

**Figure 9 cancers-14-00494-f009:**
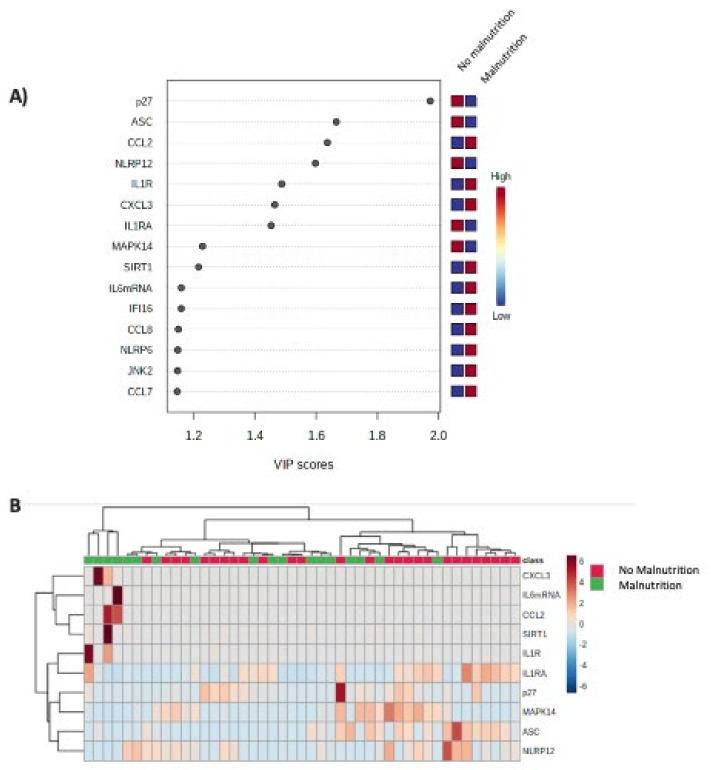
(**A**) VIP scores that summarize the contribution of the evaluated inflammasome components to discriminate patients with malnutrition (GLIM criteria); (**B**) Heatmap obtained with the expression levels of the components that better discriminate between patients with and without malnutrition.

**Table 1 cancers-14-00494-t001:** Baseline clinical characteristics of the patients. Comparison between groups based on the presence of malnutrition according to the Global Leadership Initiative on Malnutrition (GLIM) criteria or a standardized phase angle <−1.65.

Characteristics	Total (*n* = 45)	No Malnutrition (GLIM Criteria) (*n* = 27)	Malnutrition (GLIM Criteria) (*n* = 18)	p1	SPA > −1.65 (*n* = 26)	SPA < −1.65 (*n* = 18)	p2
Sex (♂/♀)	62.2/37.8% (28/17)	51.9/48.1% (14/13)	77.8/22.2% (14/4)	0.07	50/50% (13/13)	77.8/22.2% (14/18)	0.06
Age at diagnosis (years)	64.5 (61.4–73)	64 (58.7–73.8)	65 (54.8–82.4)	0.7	65 (57–76)	64 (56–78)	0.3
**Tobacco exposure**				0.9			0.5
No	21.1% (8/38)	22.7% (5/22)	18.8% (3/16)		28.6% (6/21)	12.5% (2/16)	
Active	31.1.% (4/38)	31.8% (7/22)	43.8% (7/16)		28.6% (6/21)	43.8% (7/16)	
Previous exposure	35.6% (16/38)	45.5% (10/22)	37.5% (6/16)		42.9% (9/21)	43.8% (7/16)	
**Comorbidities**							
Hypertension	37.8 (17/45)	37 (10/27)	38.9 (7/18)	0.6	38.5 (10/26)	33.3 (6/18)	0.8
Diabetes	20 (9/45)	18.5 (5/27)	22.2 (4/14)	0.5	15.4 (4/26)	22.2 (4/18)	0.7
Dyslipidemia	31.1 (14/45)	29.8 (8/27)	33.3 (6/18)	0.5	30.8 (8/26)	27.8 (5/18)	0.6
Heart disease	6.7 (3/45)	3.7 (1/27)	11.1 (2/18)	0.4	3.8 (1/26)	11.1 (2/18)	0.4
Lung disease	8.9 (4/45)	7.4 (2/27)	11.1 (2/18)	0.5	3.8 (1/26	16.7 (3/18)	0.3
Other neoplasms	11.1% (5/40)	14.8% (4/23)	5.6% (1/17)		15.4% (4/26)	5.6% (1/16)	
**Tumor localization**				0.9			0.9
Oral cavity	60% (27/45)	59.1 (16/27)	61.1% (11/18)		61.5% (16/26)	61.1% (11/18)	
Supraglottic larynx	13.3% (6/45)	11.1 (3/27)	16.7% (3/18)		7.7% (2/26)	16.7% (3/18)	
Glottic larynx	11.1% (5/45)	18.5 (5/27)	0		19.2% (5/26)	0	
Subglottic larynx	8.9% (4/45)	3.7 (1/27)	16.7% (3/18)		3.8% (1/26)	16.7% (3/18)	
Neck metastasis from unknown primary	6.6% (3/45)	7.4 (2/27)	5.6% (1/18)		7.7% (2/26)	5.6% (1/18)	
**Treatment**							
Surgery	53.3% (24/45)	55.6% (15/27)	50% (9/18)	0.5	53.8% (14/26)	50% (9/18)	0.5
Chemotherapy	55.6% (25/45)	48.1% (13/27)	66.7 (12/18)	0.2	46.2% (12/26)	66.7% (12/18)	0.2
Radiotherapy	91.1% (42/45)	88.9% (24/17)	94.4 (17/18)	0.5	88.5% (23/26)	94.4% (17/18)	0.4
**Combination therapies**							
Surgery and Radiotherapy	28.9% (13/45)	29.6% (8/27)	44.4% (8/18)	0.5	46.2% (12/26)	44.4% (8/18)	0.5
Surgery and Chemotherapy	20% (9/45)	22.2% (6/27)	16.7% (3/18)	0.5	19.2% (5/26)	16.7% (3/18)	0.6
Chemoradiotherapy	31.1% (14/45)	25.9% (7/27)	50% (9/18)	0.4	42.3% (11/26)	50% (9/18)	0.2
Surgery and Chemoradiotherapy	20% (9/45)	22.2% (6/27)	16.7% (3/18)	0.7	19.2 (5/26)	16.7 (3/18)	0.5
**Histology**				0.5			0.3
Epidermoid carcinoma	86.7 (39/45)	85.3 (23/27)	88.9 (16/18)		80.8 (21/26)	94.4 (17/18)	
Cystic adenoma	2.2 (1/45)	3.1 (1/27)	0		3.8 (1/26)	0	
Lymphoepithelioma	2.2 (1/45)	0	5.6 (1/18)		0	5.6 (1/18)	
Polymorphic adenocarcinoma	4.4 (2/45)	3.1 (1/27)	5.6 (1/18)		7.7 (2/26)	0	
Others	4.4 (2/45)	7.4 (2/27)	0		7.7 (2/26)	0	
**Cancer stage**				0.2			0.05
I	13.3 (6/45)	23.1 (6/26)	0		24 (6/26)	0	
II	6.7 (3/45)	7.7 (2/26)	5.9 (1/17)		12 (3/26)	0	
III	15.6 (7/45)	15.4 (4/26)	17.6 (3/17)		12 (3/26)	23.5 (4/17)	
IV	60 (27/45)	53.8 (14/26)	76.5 (13/17)		52 (13/26)	76.5 (13/17)	
**Symptoms**							
Weight loss (3 months)	44.4% (20/45)	37% (10/27)	55.6% (10/18)	0.2	34.9% (9/26)	55.6% (10/18)	0.1
Weight loss kg (3 months)	4 (2.6–5.6)	5 (2.3–7.4)	3 (1.6–4.8)	0.5	5 (1.8–8.1)	3.5 (2–4.5)	0.9
Weight loss (6 months)	46.7% (21/45)	44.4% (12/27)	50% (9/18)	0.5	42.3% (11/26)	55.6% (10/18)	0.3
Weight loss kg (6 months)	3.5 (2.5–7.2)	3 (1–6.7)	4 (0.6–11.2)	0.8	3.5 (0.8–7.5)	3.5 (0.9–10)	0.5
Abdominal pain	4.4% (2/45)	7.4% (2/27)	0	0.2	3.8% (1/26)	5.6% (1/18)	0.7
Nauseas/vomits	11.1% (5/45)	11.1% (3/27)	11.1% (2/18)	0.7	15.4% (4/26)	5.6% (1/18)	0.3
Diarrhea	4.4% (2/45)	7.4% (2/27)	0	0.4	7.7% (2/26)	0	0.3
Dyspnea	13.3% (6/45)	14.8% (4/27)	11.1% (2/18)	0.6	11.5% (3/26)	16.7% (3/18)	0.3
Dermatitis	28.9% (13/45)	29.6% (8/27)	27.8% (5/18)	0.5	26.9% (7/26)	27.8% (5/18)	0.5
Dysphagia	66.7% (30/45)	55.6% (15/27)	83.3% (15/18)	0.05	57.7%(15/26)	77.8% (14/18)	0.2
Mucositis	40% (18/45)	40.7% (11/27)	38.9% (7/18)	0.6	42.3% (11/26)	33.3% (6/18)	0.4
Asthenia	73.3% (33/45)	77.8% (21/27)	66.7% (12/18)	0.3	76.9% (20/26)	66.7% (12/18)	0.3
**Quality of life**							
KI	0 (0.05–2)	0 (−0.2–0.5)	2 (0.1–4.7)	0.06	0 (0–0)	2 (0.3–4)	0.02
Self-rated health score	70 (54–78)	70 (58–78)	50 (27–98)	0.3	70 (56–80)	60 (37–91)	0.4

KI: Katz Index of Independence in Activities of Daily Living; p1 refers to the comparison between non-malnutrition and malnutrition according to the GLIM criteria; p2 refers to the comparison between standardized phase angle < and >−1.65. Self-rated health score is a score between 1–100 that assesses the perceived quality of life of each patient using a visual analogue score.

**Table 2 cancers-14-00494-t002:** Morphofunctional assessment of nutritional status. Comparison between groups based on the presence of malnutrition according to the Global Leadership Initiative on Malnutrition (GLIM) criteria or a standardized phase angle < −1.65.

Characteristics	Total (*n* = 45)	No Malnutrition (GLIM Criteria) (*n* = 27)	Malnutrition (GLIM Criteria) (*n* = 18)	p1	SPA > −1.65 (*n* = 26)	SPA < −1.65 (*n* = 18)	p2
**Bioimpedance analysis**							
BMI (kg/m^2^)	24 (21.8–25.8)	23.1 (20.6–26.8)	25.6 (20.2–27.7)	0.3	**23.9 (21.8–27.3)**	**24.5 (19.3–26.7)**	**0.04**
BCMe	26.4 (23–34.2)	26.2 (21.2–30)	33.1 (18–47.7)	0.7	26.4 (22.5–30.9)	28.5 (17.8–43.2)	0.4
ECMe	16.3 (14.4–19.2)	15.5 (12.9–17.9)	18.9 (13.3–24.4)	0.7	16.3 (13.9–18.3)	17.3 (12.3–22.9)	0.9
Fat mass (%)	24.6 (21.2–32.2)	26.9 (21.5–35)	22.6 (11.4–37.5)	0.7	28.9 (21–37.2)	22.9 (14.3–34.1)	0.2
Fat mass (kg)	14.6 (12.6–20.3)	14.7 (10.7–23.1)	13.1 (8.6–23.2)	0.9	16.1 (11.6–24.8)	12.3 (8.5–21)	0.4
Lean mass (%)	71.6 (64.6–74.6)	66.9 (61.2–74.1)	73.3 (61.3–83.3)	0.4	66.3 (59.2–74.4)	73 (64.1–80.7)	0.2
Lean mass (kg)	40.1 (35.8–50.9)	39.6 (32.4–45.4)	49.4 (30.5–68.6)	0.8	40.1 (34.6–46.7)	43.4 (29–63)	0.6
Water (%)	51.4 (47.5–54.9)	49.7 (45.9–54.5)	53.4 (43.3–62)	0.7	49.7 (44.5–54.7)	53.6 (45.8–59.9)	0.3
Water (kg)	33.8 (27.6–39.2)	30 (24–36.1)	36.9 (24.7–51.5)	0.7	31.1 (25.6–37.5)	36.6 (22.8–47.7)	0.6
Bone mass (kg)	2.2 (1.9–2.7)	2.1 (1.8–2.4)	2.6 (1.7–3.5)	0.9	2.2 (1.9–2.5)	2.4 (1.6–3.3)	0.8
Anthropometric evaluation							
Abdominal circumference	90.5 (84.5–97.3)	88 (78.9–92.8)	95 (86–109)	0.9	89 (81.6–94)	93 (80.5–107.5)	0.6
Arm circumference	26 (24.9–28.2)	26 (24.1–29)	26 (23–30)	0.3	**26.5 (24.3–29.7)**	**25.5 (23.2–29.1)**	**0.05**
Calf circumference	33.5 (28.4–34.5)	33 (27.7–36)	34 (24–37)	0.2	**33.5 (28–37)**	**31 (24.6–35.8)**	**0.04**
**Muscle echography**							
Adipose tissue	0.53 (0.20–1.5)	0.69 (0.1–2.3)	0.47 (0.3–0.6)	0.3	**0.7 (−0.2–2.7)**	**0.4 (0.3–0.6)**	**0.03**
Area	1.8 (1.5–3.2)	2.15 (1.4–3.7)	1.7 (0.3–3.9)	0.3	3 (1.4–4)	1.5 (0.6–3.4)	0.2
Circunference	8.6 (6.9–9)	8.6 (6.4–9.6)	8.6 (5.7–10.2)	0.2	8.7 (7.5–9.5)	8 (5.2–9.6)	0.1
**Abdominal echography**							
Total adipose tissue	2.23 (1.8–2.7)	2 (1.7–3)	2.5 (1.12–3.11)	0.7	2.3 (1.8–3.2)	2 (1.2–2.8)	0.2
Subcutaneous adipose tissue	1.5 (1.1–1.9)	1.5 (1–2)	1.6 (0.6–2.6)	0.6	1.6 (0.9–2.1)	1.3(0.7–2.3)	0.2
Superficial subcutaneous adipose tissue	0.49 (0.4–0.7)	0.5 (0.3–0.7)	0.5 (0.1–1.1)	0.9	0.5 (0.3–0.7)	0.5 (0.2–0.9)	0.6
Deep subcutaneous adipose tissue	1 (0.8–1.3)	1.1 (0.8–1.4)	0.9 (0.6–1−3)	0.5	1.2 (0.8–1.5)	0.9 (0.5–1.2)	0.2
Preperitoneal adipose tissue	0.5 (0.3–1.2)	0.4 (0.3–0.8)	0.7 (−0.5–2.4)	0.4	0.5 (0.3–0.9)	0.6 (−0.2–2)	0.2
**Functional evaluation**							
Dynamometry (dominant arm)	18 (15–26)	19 (14–29)	17 (8–31)	0.9	22 (14–31)	17 (10–28)	0.9
Stand up test	9 (6–11)	10 (8–11)	7 (−0.08–12)	0.3	9.5 (8–10)	8 (1–12)	0.6

p1 refers to the comparison between non-malnutrition and malnutrition according to the GLIM criteria; p2 refers to the comparison between standardized phase angle < and >−1.65. BMI: body mass index; BCMe: body cell mass; ECMe: extracellular body cell mass.

**Table 3 cancers-14-00494-t003:** Biochemical analysis of the evaluated cohort. Comparison between groups based on the presence of malnutrition according to the Global Leadership Initiative on Malnutrition (GLIM) criteria or a standardized phase angle <−1.65.

Characteristics	Total (*n* = 45)	No Malnutrition (GLIM Criteria) (*n* = 27)	Malnutrition (GLIM Criteria) (*n* = 18)	p1	SPA > −1.65 (*n* =26)	SPA < −1.65 (*n* =18)	p2
**Biochemical parameters**							
Haemoglobin	13.1 (11.7–15.3)	13(11.2–13.7)	13.4 (10.3–19)	0.6	12.7(10–13)	13.4 (11.1–18.4)	0.6
Lymphocytes	910 (624–1228)	680 (296–1227)	1120 (717–1598)	0.4	810(456–1320)	1000 (965–1562)	0.9
Albumin (g/dL)	4.6 (4.3–4.8)	4.7 (4.5–4.9)	4.2 (3.9–4.7)	0.05	4.7 (4.4–4.9)	4.4 (4–4.8)	0.05
Prealbumin (mg/dL)	10.7 (17.7–26.7)	22.4 (19.3–30.6)	14.5 (9.5–27.2)	0.2	23.8(19–32)	17.2 (11.9–25)	0.3
Ferritin (mg/dL)	70.2 (36.4–121.2)	76.8 (11–160)	63.6 (10–128)	0.7	82 (6–183)	56.6 (15–110)	0.6
Transferrin (mg/dL)	260.5 (231–301)	287 (230–310)	255 (170–352)	0.03	269 (219–305)	260 (198–344)	0.06
Total cholesterol (mg/dL)	191 (175–231)	215 (175–268)	174 (162–193)	0.02	213 (160–267)	181 (152–232)	0.1
HDL cholesterol	59 (51–74)	55 (45–78)	63 (33–91)	0.7	54 (45–74)	63 (41–90)	0.9
LDL cholesterol	116 (93–139)	135 (97–163)	109 (61–129)	0.1	128 (88–167)	112(70–138)	0.2
Triglycerides	100 (85–160)	143 (71–206)	93 (67–135)	0.3	107 (50–213)	100 (72–155)	0.9
RCP	3 (1.5–10.4)	2.5 (0.8–5.2)	9.2 (−1.8–21.7)	0.03	2.4 (0.4–4.6)	7.9 (0.3–18.4)	0.04
IL-6	1.4 (0.4–11.9)	0 (−0.7–5)	2.8 (−6–3−28)	0.01	0 (−1.5–5.6)	2.6(−4–22.8)	0.007
Zinc (mg/dL)	70.6 (66.7–90.3)	70 (58.6–85.8)	91.7 (60–113)	0.2	67 (59–81)	93 (68–108)	0.08
Serotonin	127.5 (92–209)	197 (78–262)	117 (20–226)	0.9	159 (60–229)	126 (41–272)	0.5
Vitamin D	17 (13–28)	17 (8–36)	17 (11–27)	0.3	22 (9–40)	16 (9–24)	0.5

p1 refers to the comparison between non-malnutrition and malnutrition according to the GLIM criteria; p2 refers to the comparison between standardized phase angle < and >−1.65. RCP: reactive C protein; IL: interleucine.

## Data Availability

Not applicable.

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
