# Peer review of "Morphofunctional and Molecular Assessment of Nutritional Status in Head and Neck Cancer Patients Undergoing Systemic Treatment: Role of Inflammasome in Clinical Nutrition"

_cancers, 2022, doi:10.3390/cancers14030494_

Round 1

Reviewer 1 Report

This is an interesting study about the role of inflammasome in nutrition of head and neck cancer patients.

The paper is well written. However, some issues remain.

Simple summary, conclusions, author contributions, institutional review board statement, informed consent statement, and data availability statement are absent.

Since the authors evaluated patients with head and neck cancers who were receiving systemic treatment, it should be stated in the title.

Patients with unknown tumor localization should be excluded from the study if the data is lacking or included if the authors meant neck metastasis from unknown primary.

In table 1, neck tumor localization must be specified.

Treatments must be better described: did how many patients undergo exclusive chemoradiotherapy and how surgery plus adjuvant treatment?

Please describe what is “self-rated health score” in table 1.

At which timing were clinical evaluations and sample collection performed?

All the acronyms of tables 2 and 3 must be explained.

Author Response

Point-by-point response to the Reviewer’s comments

We sincerely thank the Editor for the interest in our patient cohort and results. Following your suggestions, we re-evaluated our manuscript and focused on the comments of the Reviewers. We sincerely thank the Reviewers for their constructive comments, which we found very helpful towards improving the quality of our study. Accordingly, specific changes have been made in the manuscript, based on these comments, as it is described in detail below in a point-by-point description of the changes introduced, and on how Reviewer’s concerns were addressed. Changes within the manuscript are indicated in red.

Responses to Reviewer #1

REVIEWER:

 This is an interesting study about the role of inflammasome in nutrition of head and neck cancer patients.The paper is well written. However, some issues remain.

Reviewer: Simple summary, conclusions, author contributions, institutional review board statement, informed consent statement, and data availability statement are absent.

Authors: Simple summary, conclusions, author contributions, institutional review board statement, informed consent statement, and data availability have been included on the revised version of our manuscript.

Reviewer: Since the authors evaluated patients with head and neck cancers who were receiving systemic treatment, it should be stated in the title.

Authors: As the reviewer suggested, the title has been modified.

Reviewer: Patients with unknown tumor localization should be excluded from the study if the data is lacking or included if the authors meant neck metastasis from unknown primary.

Authors: Clinical history of the patients has been reviewed and corresponded to neck tumors.

Reviewer: In table 1, neck tumor localization must be specified.

Authors: Neck tumors have been specified in the revised table 1

Reviewer: Treatments must be better described: did how many patients undergo exclusive chemoradiotherapy and how surgery plus adjuvant treatment?

Authors: Additional information about combined treatment was included in Table 1 of the revised version of our manuscript.

Reviewer: Please describe what is “self-rated health score” in table 1.

Authors: As the reviewer suggested, a description of the self-rated health score was included in the legend of table 1.

Reviewer: At which timing were clinical evaluations and sample collection performed?

Authors: Patients were evaluated when referred to the Endocrinology Service in our hospital, this information was included in the materials and methods section of the revised version of our manuscript.

Reviewer: All the acronyms of tables 2 and 3 must be explained.

Authors: As the reviewer suggested, the acronyms of tables 2 and 3 were explained in the revised version of our manuscript.

Reviewer 2 Report

Graphics in this article could be improved for legibility. Also, see my comments in the attached pdf file.

Author Response

Point-by-point response to the Reviewer’s comments

We sincerely thank the Editor for the interest in our patient cohort and results. Following your suggestions, we re-evaluated our manuscript and focused on the comments of the Reviewers. We sincerely thank the Reviewers for their constructive comments, which we found very helpful towards improving the quality of our study. Accordingly, specific changes have been made in the manuscript, based on these comments, as it is described in detail below in a point-by-point description of the changes introduced, and on how Reviewer’s concerns were addressed. Changes within the manuscript are indicated in red.

Responses to Reviewer #2

REVIEWER:

Reviewer: Graphics in this article could be improved for legibility. Also, see my comments in the attached pdf file.

Authors: We thank the reviewer for the comments, which have been included in the revised version of our manuscript.

Reviewer 3 Report

The Authors explored epidemiological/clinical/anthropometric/biochemical factors in 45 patients affected with head and neck cancer and treated with surgery, radiotherapy, systemic agents. Serum RCP, IL6 and molecular expression of inflammasome-components  and inflammatory-associated factors were evaluated in peripheral-blood mononuclear-cells (PBMCs). The article is of interest.

  • Please provide a Simple Summary which is not present in the current version of the manuscript.
  • ‘’To perform a novel morfofunctional nutritional evaluation and explore changes in inflammasome-machinery components in patients with head and neck cancer who are undergoing systemic treatment. Epidemiological/clinical/anthropometric/biochemical evaluation of forty-five patients was performed’’. Shouldn’t the 2 sentences be combined into a single one?
  • Abstract Line 25. What is RCP. Use full text before the acronym.
  • Line 49. Also radiotherapy may worsen nutritional conditions. Please add.
  • Line 92. Patients also received radiotherapy. Please state it.
  • Please remove Conclusions if not necessary or add it up with text if deemed appropriate
  • Please feel with text Authors’ contribution, Institutional Review Board Statement, Informed Consent Statement and Data Availability Statement.

Author Response

Point-by-point response to the Reviewer’s comments

We sincerely thank the Editor for the interest in our patient cohort and results. Following your suggestions, we re-evaluated our manuscript and focused on the comments of the Reviewers. We sincerely thank the Reviewers for their constructive comments, which we found very helpful towards improving the quality of our study. Accordingly, specific changes have been made in the manuscript, based on these comments, as it is described in detail below in a point-by-point description of the changes introduced, and on how Reviewer’s concerns were addressed. Changes within the manuscript are indicated in red.

Responses to Reviewer #3

REVIEWER:

The Authors explored epidemiological/clinical/anthropometric/biochemical factors in 45 patients affected with head and neck cancer and treated with surgery, radiotherapy, systemic agents. Serum RCP, IL6 and molecular expression of inflammasome-components  and inflammatory-associated factors were evaluated in peripheral-blood mononuclear-cells (PBMCs). The article is of interest.

Reviewer: Please provide a Simple Summary which is not present in the current version of the manuscript.

Authors: A simple summary was included in the revised version of our manuscript as the reviewer suggested.

Reviewer: ‘’To perform a novel morfofunctional nutritional evaluation and explore changes in inflammasome-machinery components in patients with head and neck cancer who are undergoing systemic treatment. Epidemiological/clinical/anthropometric/biochemical evaluation of forty-five patients was performed’’. Shouldn’t the 2 sentences be combined into a single one?

Authors: As suggested, we corrected the referred sentences of the abstract.

Reviewer: Abstract Line 25. What is RCP. Use full text before the acronym.

Authors: The acronym was explained in the revised version of our manuscript.

Reviewer: Line 49. Also radiotherapy may worsen nutritional conditions. Please add.

Authors: As the reviewer suggested, the influence of systemic treatment on nutritional conditions was included.

Reviewer: Line 92. Patients also received radiotherapy. Please state it.

Authors: As the reviewer suggested, in the materials and methods section we explained that patients received radiotherapy.

Reviewer: Please remove Conclusions if not necessary or add it up with text if deemed appropriate

Authors: We included a conclusions section in the revised version of our manuscript.

Reviewer: Please feel with text Authors’ contribution, Institutional Review Board Statement, Informed Consent Statement and Data Availability Statement.

Authors: Authors’ contribution, Institutional Review Board Statement, Informed Consent Statement and Data Availability Statement have been included in the revised version of our manuscript.

Reviewer 4 Report

Dear Editor

I carefully reviewed the manuscript ID-1479895, entitled as” Morphofunctional and Molecular Assessment of Nutritional Status in Head and Neck Cancer Patients: Role of Inflammasome in Clinical Nutrition " submitted by Soraya León Idougourram. et al. The authors prospectively analyzed the nutrition and inflammation status of 45 patients using several methods including clinic records, anthropometric measurements, bioimpedance analysis (BIA), serum nutritional and inflammatory markers and the molecular analysis of inflammasome. They found several interesting and statistically significant correlations between the abovementioned factors. Further the authors showed specific components of the inflammasome machinery in combination with serum inflammation markers might play an important role as indicators for predicting malnutrition and related comorbidities. In overall, the authors did a hard work to show their correlations and implications. Nonetheless, there are several concerns that make these results less convincing.

Major concerns:

  1. Different cancer stages have varied nutrition and inflammation condition. There is no information in TNM stage of patients, including T status, N involvement and M, metastasis. There is no histological grade differentiation, alcohol exposure, comorbid status and performance status, either. These factors had impacts on nutrition and inflammation conditions. If the authors try to develop a tool to complement the currently known nutritional assessment, it is better to provide to show that the analysis of this manuscript is better than these basic demographic variables in the nutrition assessment.
  2. Were all participants’ data collected before treatment? Was this aim of the study to assess pretreated nutrition and inflammation status of patients with HNC? If yes, therapy in Table 1 may not be necessarily provided.
  3. Several abbreviations were shown at the first appearance with no whole spelling. This make the readers to follow.
  4. The methodology is full of speculations and assumptions. For instance, authors adapted Reference 18 (Berger, J., et al., Rectus femoris (RF) ultrasound for the assessment of muscle mass in older people. Arch Gerontol Geriatr, 2015. 61(1): 554 p. 33-8). In Berger’s study, they applied “General Electric Logiq ultrasonogapher” to healthy people, not cancer patients. Also, they found male gender with old age have difference from young age and female gender. In Table 1, the distributions of age and gender distribution really raised a concern in the current data reliability. Moreover, the current analysis used Midray Z50 Ultrasound, which is different from Berger’s study. The inter-instrumental and inter-personal variations in assessment should be clearly and precisely evaluated. Similarly, authors adapted Reference 10 to assess fat tissue condition (Hamagawa, K., et al., Abdominal visceral fat thickness measured by ultrasonography predicts the presence and severity of coronary artery Ultrasound Med Biol, 2010. 36(11): p. 1769-75). This application is also highly concerned because Hamagawa’s result is for coronary artery disease, not cancer patients.
  5. Is the cutoff for SPA 1.65 or -1.65? It is different Table 1, text and Table 2. Different cutoff may be generated from varied BIA machines and cancer types. It may not be able to apply one value to apply the other. It is better to set up by the current analysis.
  6. Is there correlation between the parameters of BIA data and ultrasound records including muscle and fat data in this study?
  7. The current data tend to correlation analysis, not complimentary.
  8. English grammars need to be polished.

Author Response

Point-by-point response to the Reviewer’s comments

We sincerely thank the Editor for the interest in our patient cohort and results. Following your suggestions, we re-evaluated our manuscript and focused on the comments of the Reviewers. We sincerely thank the Reviewers for their constructive comments, which we found very helpful towards improving the quality of our study. Accordingly, specific changes have been made in the manuscript, based on these comments, as it is described in detail below in a point-by-point description of the changes introduced, and on how Reviewer’s concerns were addressed. Changes within the manuscript are indicated in red.

Responses to Reviewer #4

REVIEWER:

Dear Editor

I carefully reviewed the manuscript ID-1479895, entitled as” Morphofunctional and Molecular Assessment of Nutritional Status in Head and Neck Cancer Patients: Role of Inflammasome in Clinical Nutrition " submitted by Soraya León Idougourram. et al. The authors prospectively analyzed the nutrition and inflammation status of 45 patients using several methods including clinic records, anthropometric measurements, bioimpedance analysis (BIA), serum nutritional and inflammatory markers and the molecular analysis of inflammasome. They found several interesting and statistically significant correlations between the abovementioned factors. Further the authors showed specific components of the inflammasome machinery in combination with serum inflammation markers might play an important role as indicators for predicting malnutrition and related comorbidities. In overall, the authors did a hard work to show their correlations and implications. Nonetheless, there are several concerns that make these results less convincing.

Major concerns:

Reviewer: Different cancer stages have varied nutrition and inflammation condition. There is no information in TNM stage of patients, including T status, N involvement and M, metastasis. There is no histological grade differentiation, alcohol exposure, comorbid status and performance status, either. These factors had impacts on nutrition and inflammation conditions. If the authors try to develop a tool to complement the currently known nutritional assessment, it is better to provide to show that the analysis of this manuscript is better than these basic demographic variables in the nutrition assessment.

Authors: The aim of the study was to evaluate the clinical status of patients with head and neck tumors undergoing systemic treatment. In our center and according to the clinical practice guidelines, nutritional evaluation and support should be provided to all patients with cancer with positive initial screening independently of TNM or histological differentiation. Since these factors may introduce additional confounding variables, we did not include them in the analysis and consequently in the manuscript. Performance status was evaluated using the katz index (Table 1)

Reviewer: Were all participants’ data collected before treatment? Was this aim of the study to assess pretreated nutrition and inflammation status of patients with HNC? If yes, therapy in Table 1 may not be necessarily provided.

Authors: patients were evaluated when receiving systemic treatment. We better specified this matter In the Materials and Methods section of the revised version of our manuscript.

Reviewer: Several abbreviations were shown at the first appearance with no whole spelling. This make the readers to follow.

Authors: As the reviewer suggested, some acronyms have been explained in the revised version of our manuscript.

Reviewer: The methodology is full of speculations and assumptions. For instance, authors adapted Reference 18 (Berger, J., et al., Rectus femoris (RF) ultrasound for the assessment of muscle mass in older people. Arch Gerontol Geriatr, 2015. 61(1): 554 p. 33-8). In Berger’s study, they applied “General Electric Logiq ultrasonogapher” to healthy people, not cancer patients. Also, they found male gender with old age have difference from young age and female gender. In Table 1, the distributions of age and gender distribution really raised a concern in the current data reliability. Moreover, the current analysis used Midray Z50 Ultrasound, which is different from Berger’s study. The inter-instrumental and inter-personal variations in assessment should be clearly and precisely evaluated. Similarly, authors adapted Reference 10 to assess fat tissue condition (Hamagawa, K., et al., Abdominal visceral fat thickness measured by ultrasonography predicts the presence and severity of coronary artery Ultrasound Med Biol, 2010. 36(11): p. 1769-75). This application is also highly concerned because Hamagawa’s result is for coronary artery disease, not cancer patients.

Authors: The use of ultrasound for evaluating muscle and fat mass is a novel technique in clinical nutrition. Currently in Spain, the Spanish Society of Endocrinology is trying to unify this method in order to improve the reliability of these measurements. Moreover, all the Society members that work with clinical nutrition are receiving the same training; specifically, the techniques of Berger and Hamagawa have been adapted. These changes in the nutritional evaluation have been recently published by Garcia Almeida (Morphofunctional assessment of patient s nutritional status: a global approach. Nutr Hosp, 2021. 38(3): p. 592-600), who is the responsible in the Society. We agree with reviewer regarding the inter-instrumental and inter-personal variations, to this aim, we specifically referred to this matter in the Discussion section of the revised version of our manuscript (lines 407-408).

Reviewer: Is the cutoff for SPA 1.65 or -1.65? It is different Table 1, text and Table 2. Different cutoff may be generated from varied BIA machines and cancer types. It may not be able to apply one value to apply the other. It is better to set up by the current analysis.

Authors: Indeed, the cutoff level for SPA was -1.65. We corrected this mistake in the revised version of our manuscript.

Reviewer: Is there correlation between the parameters of BIA data and ultrasound records including muscle and fat data in this study? The current data tend to correlation analysis, not complimentary.

Authors: BIA and ultrasound measurements were correlated, due to the large number of results, only significant results were presented. Following the reviewer suggestion, we specified this matter in the revised version of our manuscript (lines 229-230).

Reviewer: English grammars need to be polished.

Authors: as the reviewer suggested, a native speaker revised the final version of our manuscript.

Round 2

Reviewer 1 Report

"Cervical localization" is not enough. Please better specify. Did you mean "larynx" or "neck metastasis from unknown primary"?

About treatment modalities in table 1, the sum of the three strategies is over 100%. It is not possible. Please correct.

Is self-rated health score similar to a visual analogue score? Please better describe.

Concerning timing of evaluations, the authors must specify a time interval.

Author Response

Point-by-point response to the Reviewer’s comments

We sincerely thank the Editor for the interest in our patient cohort and results. Following your suggestions, we re-evaluated our manuscript and focused on the comments of the Reviewers. We sincerely thank the Reviewers for their constructive comments, which we found very helpful towards improving the quality of our study. Accordingly, specific changes have been made in the manuscript, based on these comments, as it is described in detail below in a point-by-point description of the changes introduced, and on how Reviewer’s concerns were addressed. Changes within the manuscript are indicated in red.

Responses to Reviewer #1

REVIEWER:

Reviewer: "Cervical localization" is not enough. Please better specify. Did you mean "larynx" or "neck metastasis from unknown primary"?

Authors:  As the reviewer suggested we specified cervical location as neck metastasis from unknown primary in the revised version of our manuscript

Reviewer: About treatment modalities in table 1, the sum of the three strategies is over 100%. It is not possible. Please correct.

Authors:  We corrected this mistake in table 1

Reviewer: Is self-rated health score similar to a visual analogue score? Please better describe.

Authors:  Yes, we specified this in the revised version of our manuscript.

Reviewer: Concerning timing of evaluations, the authors must specify a time interval.

Authors:  we specified the time interval in the revised version of our manuscript

Reviewer 4 Report

Dear Editor

I carefully reviewed the revised manuscript ID-1479895, entitled as” Morphofunctional and Molecular Assessment of Nutritional Status in Head and Neck Cancer Patients: Role of Inflammasome in Clinical Nutrition " submitted by Soraya León Idougourram. et al. The authors answered all my concerns, except the effect of tumor stage, comorbid status and histologic grade on the currently presented data. These covariates are basic and critical to all cancer studies, particularly in inflammation and nutrition fields. Although authors presented a sophisticated analysis and the result sounded reasonable, it is still difficult to convince clinical healthcare professional participating in cancer patient care to accept such an analysis without adjustment of these covariates. I deeply concern the author reply “Since these factors may introduce additional confounding variables, we did not include them in the analysis and consequently in the manuscript”. These covariates are not confounding but really essential to cancer study. If the authors could offer a better explanation in this regard, the manuscript should be acceptable.

The other minor question in “tumor location” of Table 1, what does “cervical location” mean? It is not appropriate to use this term for head and neck cancer. Please correct it following AJCC 8th version for tumor stage.

Author Response

Responses to Reviewer #2

REVIEWER:

Reviewer: Dear Editor, I carefully reviewed the revised manuscript ID-1479895, entitled as” Morphofunctional and Molecular Assessment of Nutritional Status in Head and Neck Cancer Patients: Role of Inflammasome in Clinical Nutrition " submitted by Soraya León Idougourram. et al. The authors answered all my concerns, except the effect of tumor stage, comorbid status and histologic grade on the currently presented data. These covariates are basic and critical to all cancer studies, particularly in inflammation and nutrition fields. Although authors presented a sophisticated analysis and the result sounded reasonable, it is still difficult to convince clinical healthcare professional participating in cancer patient care to accept such an analysis without adjustment of these covariates. I deeply concern the author reply “Since these factors may introduce additional confounding variables, we did not include them in the analysis and consequently in the manuscript”. These covariates are not confounding but really essential to cancer study. If the authors could offer a better explanation in this regard, the manuscript should be acceptable.

Authors:  As the reviewer suggested, information about histology, tumor stage and comorbidities was included in table 1 of the revised version of our manuscript.

Reviewer: The other minor question in “tumor location” of Table 1, what does “cervical location” mean? It is not appropriate to use this term for head and neck cancer. Please correct it following AJCC 8thversion for tumor stage.

Authors:  As the reviewer suggested we specified cervical location as neck metastasis from unknown primary in the revised version of our manuscript .